# Electrolysis of a molten semiconductor

Huayi Yin[1], Brice Chung[1] & Donald R. Sadoway[1]

Metals cannot be extracted by electrolysis of transition-metal sulfides because as liquids they are semiconductors, which exhibit high levels of electronic conduction and metal dissolution. Herein by introduction of a distinct secondary electrolyte, we reveal a high-throughput electro-desulfurization process that directly converts semiconducting molten stibnite ($Sb_2S_3$) into pure (99.9%) liquid antimony and sulfur vapour. At the bottom of the cell liquid antimony pools beneath cathodically polarized molten stibnite. At the top of the cell sulfur issues from a carbon anode immersed in an immiscible secondary molten salt electrolyte disposed above molten stibnite, thereby blocking electronic shorting across the cell. As opposed to conventional extraction practices, direct sulfide electrolysis completely avoids generation of problematic fugitive emissions ($CO_2$, CO and $SO_2$), significantly reduces energy consumption, increases productivity in a single-step process (lower capital and operating costs) and is broadly applicable to a host of electronically conductive transition-metal chalcogenides.

[1] Department of Materials Science and Engineering, Massachusetts Institute of Technology, Cambridge, Massachusetts 02139-4307, USA. Correspondence and requests for materials should be addressed to D.R.S. (email: dsadoway@mit.edu).

Direct electrochemical reduction of ores can improve metal recovery by simplifying the process, reducing energy consumption, as well as capital and operating costs, and offering cleaner, sustainable extraction pathways. The earliest example is aluminium production, thanks to the invention in 1886 of the Hall–Héroult process, which displaced the once conventional pyrometallurgical route and turned aluminium from a precious metal costing more than silver into a common structural material[1]. More recently, as an alternative to the established pyrometallurgical processes for titanium metal production (Hunter and Kroll), Fray, Farthing and Chen presented the idea of direct electrolytic reduction of solid titanium dioxide, the so-called FFC process[2]. On the horizon is molten oxide electrolysis, which has been shown by Sadoway et al. to produce liquid iron and by-product oxygen[3,4]. Analogous progress with electrolytic reduction of molten sulfides has been obstructed by their high melting temperature, which results in high vapour pressure, the lack of a practical inert anode, and the high degree of electronic conductivity and metal solubility in these melts[5–10], which results in unacceptably high levels of cell shorting with attendant high energy consumption and low Faradaic efficiency[11,12]. Attempts to suppress electronic shorting across the cell typically resort to diluting the semiconducting compound in an ionic melt[13,14] with the intention of decreasing the electronic conductivity of the feedstock by lowering the concentration of solvated electrons or by creating trapping centres that lower electronic mobility[8]. Yet dilution unfavourably decreases mass transport with resultant loss in cell productivity. In some instances candidate additives can compete for electro-extraction with the element of interest and at high current density co-deposit, which renders the product unmarketable without further purification.

In a radical departure from ionic dilution, here we instead adopt a different strategy, one that does not attempt to reduce electronic conduction of the feedstock but rather inhibits electron access to one of the cell's electrodes by deployment of a discrete electron-blocking secondary molten electrolyte so as to enable direct metal recovery from ores that were previously deemed electrochemically irreducible. By way of example we demonstrate the production of high-purity liquid antimony via direct electrolysis of the molten semiconductor, antimony sulfide ($Sb_2S_3$), derived from its predominant ore (mineral stibnite), in stark contrast to the more complicated traditional pyrometallurgical and hydrometallurgical extraction pathways (Supplementary Note 1). Furthermore, since all reactants and products are fluid, we have the scalable conditions for a simple, continuous and high-throughput industrial process. Easy metal recovery is further enabled by the density ranking of Sb, $Sb_2S_3$ and the secondary molten halide electrolyte, which self-segregate into three immiscible layers.

## Results

**Layout of electrolysis cell.** As seen in Fig. 1, at the bottom of the cell higher density liquid antimony pools beneath cathodically polarized molten stibnite. At the top of the cell, sulfur issues from a carbon anode immersed in the electron-blocking secondary halide electrolyte, a separate layer distinct from the underlying molten stibnite feedstock. We observe high metal production rates and total conversion of the feedstock. Complete fluidity enables ready collection of products.

**Selection of electrolyte.** According to the phase diagrams[15,16], $Na_2S$ and $K_2S$ are highly soluble in molten NaCl–KCl. This was confirmed experimentally in a transparent fused quartz cell charged with molten $NaCl_{50.6mol\%}$–$KCl_{49.1\%}$ and 2 wt% $Na_2S$. As

shown in Supplementary Fig. 1, molten NaCl–KCl–$Na_2S$ was observed to be uniform, red in colour and devoid of any precipitates, indicative of a single-phase liquid. In contrast, $Sb_2S_3$ has only limited solubility in molten NaCl–KCl–$Na_2S$ at 700 °C (ref. 17). On this basis, we chose $Na_2S$ dissolved in molten NaCl–KCl to serve as a secondary electrolyte that conducts sulfide ion while obstructing the flow of electrons. Halides have previously been used in the FFC process for the reduction of solid-state feedstocks, including sulfides[18,19]. For example, solid $MoS_2$ was electrolysed in molten $CaCl_2$ to produce Mo (ref. 20). However, it was found that slow $S^{2-}$ transport throttled productivity and caused CaS contamination. In another instance, tungsten powder was prepared by electrochemical reduction of solid $WS_2$ in molten NaCl–KCl (ref. 21). While these results demonstrate electrolytic reduction of a sulfide compound at moderate temperatures, conversion of solid feedstock into solid metal product is necessarily confined to three-phase conductor/insulator/electrolyte reaction sites, which impedes throughput rendering the process challenging at industrial scale for its low space-time yield[22,23]. Herein we overcome these limitations by resorting to an all-fluid system: molten semiconducting feedstock converting to liquid metal product and sulfur vapour by-product, the latter mediated by a molten halide electrolyte doped to render it a sulfide-ion conductor. While Hoar and Ward demonstrated the electrolysis of molten copper sulfide in complicated bicameral laboratory cells containing a cathode and an anode both composed of molten copper sulfide and connected by molten barium chloride chosen for its low vapour pressure at 1,150 °C (ref. 24), the absence of alkali sulfide dissolved in the barium chloride led to generation of cuprous ions at the anode with attendant precipitation of $Cu_2S$. Indeed, the authors admit that while twin copper sulfide electrodes can be made to work in small, laboratory-scale cells, 'large-scale cells would obviously present formidable development problems, not the least being methods for feeding molten white metal into the electrode compartments.'

The cell proposed herein obviates the need for bicameral feeding, and since the molten antimony sulfide naturally lies

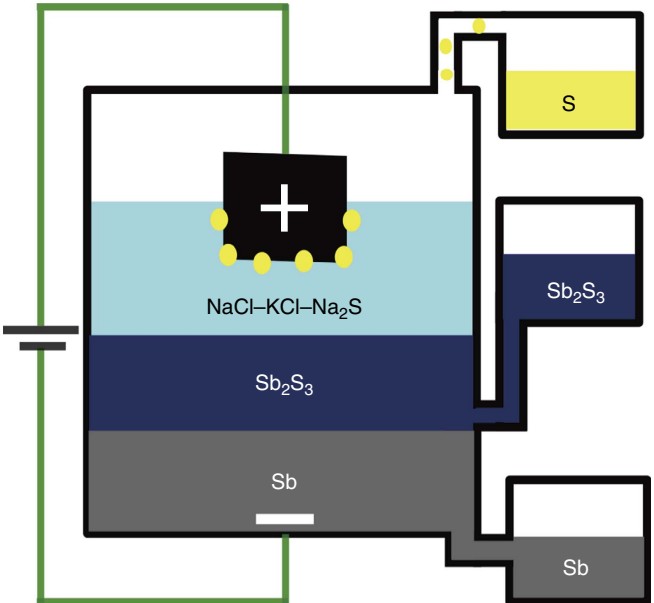

**Figure 1 | Schematic of three-layered electrolysis cell.** Schematic illustrating the use of an electron-blocking secondary electrolyte for direct electrolytic conversion of molten semiconducting $Sb_2S_3$ into liquid Sb and sulfur vapour.

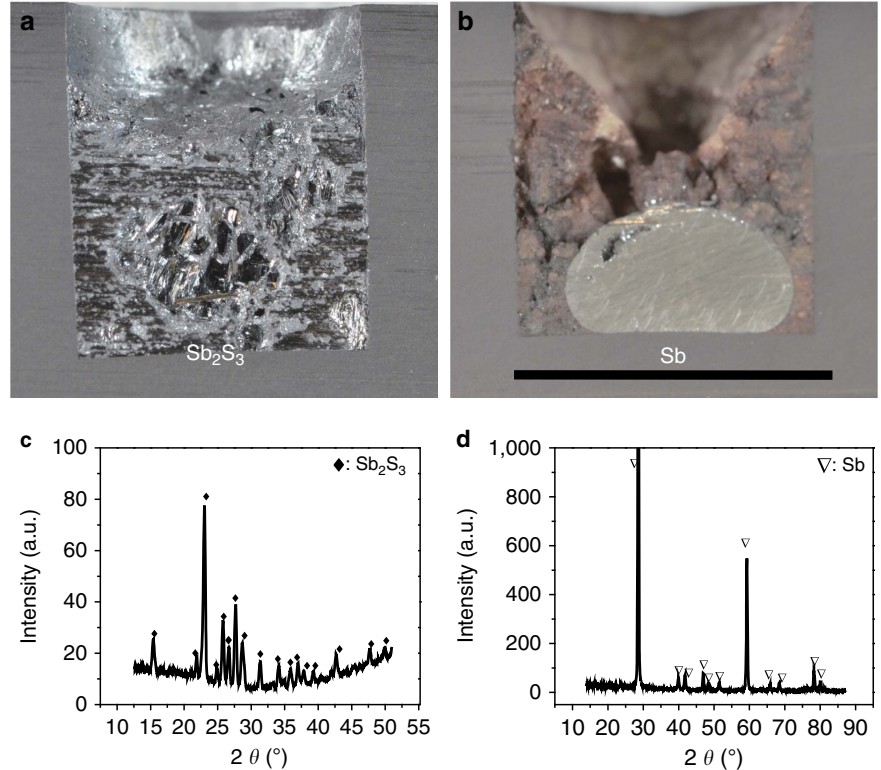

**Figure 2 | Image and X-ray diffraction analysis of cathode product.** Cross-sectional images of (**a**) the $Sb_2S_3$ electrode and (**b**) the electrolytic Sb. X-ray diffraction patterns of (**c**) the $Sb_2S_3$ feedstock and (**d**) the electrolytic Sb product. Scale bar, 1 cm.

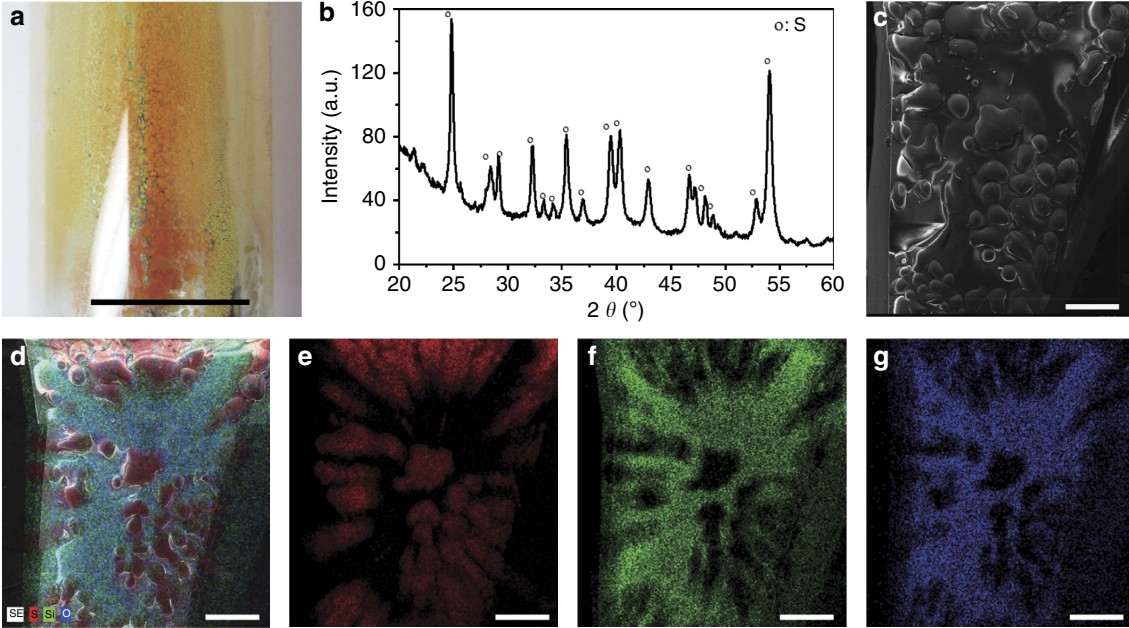

**Figure 3 | Analysis of anode product.** (**a**) Image of the anode product generated on the side wall of the fused quartz cell. (**b**) X-ray diffraction patterns of the yellow anode product (orthorhombic sulfur in the database: PDF No. 00-008-0247). (**c**) Scanning electron microscopy image of the yellow anode product. (**d–g**) EDS element mappings of the yellow anode product: (**d**) the overlap mapping of sulfur, silicon and oxygen; (**e**) the red is sulfur; (**f**) the green is silicon; and (**g**) and the blue is oxygen. The scale of **a** is 2 cm and **c–f** is 500 μm.

between antimony metal and the molten chloride electrolyte, periodic charging of feedstock and periodic tapping of metal product is more straightforward. No antimony ions are introduced into the molten chloride electrolyte, which for us acts as a sulfide ion conductor and pathway to the anode. Put another way, this cell exploits the electro-desulfurization concept of FFC while achieving scalable throughput thanks to the liquidity of both feedstock and product metal.

**Electrolysis.** Here in a three-electrode set-up at 700 °C, we demonstrate the conversion of molten $Sb_2S_3$ to liquid Sb and sulfur vapour by the action of electric current at constant applied potential. As shown in Fig. 2a, a sample of 1 g $Sb_2S_3$ (molten $Sb_2S_3$ layer of thickness ∼1 cm) was entirely electrolysed in <49 min. The half reactions can be written as,

$$Cathode\ (in\ molten\ semiconductor):$$
$$Sb_2S_3(\ell) + 6\,e^- = 2\,Sb(\ell) + 3\,S^{2-} \tag{1}$$

$$Anode\ (in\ molten\ secondary\ electrolyte):$$
$$3\,S^{2-} = \frac{3}{x}\,S_x(g) + 6\,e^- \tag{2}$$

After operation, the cell was sectioned, and a metal bead was observed at the bottom of the graphite electrode (Fig. 2b), confirming the production of high-density liquid metal at the cathode. Remarkably, $Sb_2S_3$ was fully converted to elemental Sb (X-ray diffraction in Fig. 2c,d), and the purity of the extracted metal exceeded 99.9% (energy-dispersive X-ray spectroscopy (EDS) in Supplementary Fig. 2). Significantly, no elemental sodium, potassium, nor halide or sulfide compounds were found

in the metal product, suggesting that the electrolytically extracted Sb needs no further treatment. As is the case with all electrolytic reduction operations, high-purity product is predicated on high-purity feedstock; no metal refining can be expected in cells designed for primary extraction. In parallel, in the vicinity of the anode a yellow condensate was observed on the fused quartz cell wall (Fig. 3a) and identified by EDS (Supplementary Fig. 3) and X-ray diffraction (Fig. 3b) to be high-purity, orthorhombic sulfur. This is consistent with the condensation of sulfur vapour following its evolution on the graphite anode. In Fig. 3e–g the presence of elemental Si and O corresponds to the fused quartz substrate on which the sulfur had deposited.

To determine the operational envelope (extraction rate and cell voltage) relative to the secondary electrolyte's electrochemical window, the potential of the anode (counter electrode) was monitored *in situ* during potentiostatic electrolysis. Sulfur evolution is expected to occur at 1.55 V (versus $Na^+/Na$) while undesirable chlorine evolution is expected to occur at potentials above 3.3 V (versus $Na^+/Na$, Supplementary Table 1), which in our experimental set-up (Supplementary Fig. 4) is achieved at a current density of 550 mA cm$^{-2}$. Accordingly, galvanostatic electrolysis was conducted at 500 mA cm$^{-2}$. As shown in Fig. 4a, in the first 10 s, a sharp rise in cell voltage was observed. This is principally attributed to polarization at the anode (increase in potential from 2.2 to 2.8 V versus $Na^+/Na$) on which sulfur vapour evolves. At the cathode, polarization is minimal, consistent with fast charge-transfer kinetics and rapid mass transport associated with electrodeposition of liquid metal from molten salt. Over time, as feedstock is depleted, cathode potential predictably decreases (becomes more negative) and cell voltage increases.

After galvanostatic electrolysis, a bead of high-purity Sb was observed at the bottom of the graphite container (Fig. 4b). On visual inspection, the anodic graphite rod revealed no signs of erosion despite service for a complete week (Supplementary Fig. 5). The voltage recorded at the anode during galvanostatic electrolysis is in agreement with cyclic voltammetry on graphite showing that oxidation occurs at potentials exceeding 2.2 V (Supplementary Fig. 6). By comparison of the mass of the electrolytic Sb to the integrated current during the course of galvanostatic electrolysis at the high constant current density of 500 mA cm$^{-2}$, the Faradaic current efficiency is determined to be 88% with an energy consumption of 1.5 kWh per kg Sb.

## Discussion

Here in a cell comprising an electron-blocking secondary molten NaCl–KCl–Na$_2$S electrolyte, efficient direct electrolysis of molten semiconducting $Sb_2S_3$ has been shown to produce in a single-step high-purity liquid Sb and S vapour at a high rate of 500 mA cm$^{-2}$ without fugitive gas emissions, starting with feedstock derived from antimony's predominant ore, and achieving a low energy consumption of 1.5 kWh per kg Sb. It has not escaped our notice that the immiscible secondary electrolyte approach could well be adapted for use with other molten semiconductors, not only sulfides, but even transition-metal oxides[25]. The combination of eutectic mixes of ores to decrease melting temperature of the charge, of alloying of metals to allow for liquid metal collection and of design of specific secondary electrolytes to vitiate electronic shorting while facilitating transport of the chalcogenide ion, could pave the way for the recovery of other elements by sustainable, modern electrometallurgical means. The key to exploitation of this new approach is the dual recognition of the semiconducting nature of molten transition-metal chalcogenides and their immiscibility with both liquid transition metals and molten alkali-metal halides, which can be doped to be chalcogenide-ion conductors.

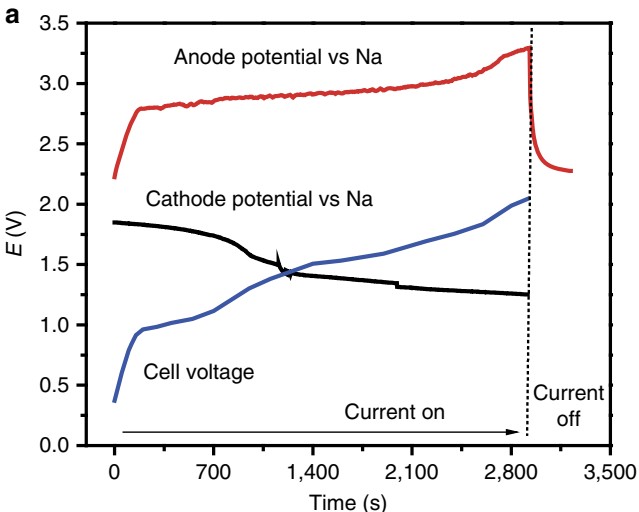

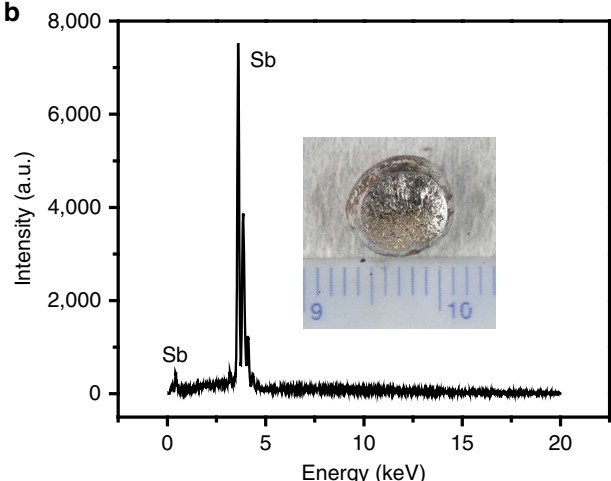

**Figure 4 | Voltage time traces and cathode product of galvanostatic electrolysis. (a)** Cathode, anode and cell voltage time traces during galvanostatic electrolysis at 500 mA cm$^{-2}$. **(b)** EDS spectrum of the obtained Sb; inset is the image of the electrolytic bead of Sb. The scale bar of the inset is 1.5 cm.

## Methods

**Electrolysis set-up and materials.** A three-electrode set-up was sealed in a stainless-steel test vessel heated in a tube furnace. The working electrode consisted of a graphite cup (inner diameter (ID): 1 cm) containing 1 g $Sb_2S_3$ pre-melted by an induction smelter in an argon-filled glove box. The Ag/AgCl reference electrode was fabricated by loading 1.5 g $LiCl_{59.2 \text{ mol}\%}$–$KCl_{40.8\%}$ (99.9% purity) with 2 wt% AgCl into a closed-one-end mullite tube (ID = 4 mm), and inserting a Ag wire to serve as the current lead. The counter electrode was composed of a graphite rod (6 mm outer diameter, >99.999%) connected to a tungsten wire. The electrolyte was composed of 500 g $NaCl_{50.6 \text{mol}\%}$–$KCl_{49.1\%}$ (99.9% purity) with 2 wt% $Na_2S$ contained in an alumina crucible (ID 6 cm, height 12 cm). In an inert atmosphere the cell was assembled and transferred to a stainless-steel test vessel. The assembly was then initially held under vacuum at 250 °C for 12 h to remove residual moisture. The test vessel was then refilled with high-purity Ar gas and kept under constant flow while the temperature was ramped up to 700 °C.

**Electrochemistry and characterization.** Cyclic voltammetry was initially conducted with a tungsten working electrode to calibrate the Ag/AgCl reference electrode. Electrochemical behaviour of graphite was characterized with a graphite rod working electrode and the aforementioned reference and counter electrodes in the NaCl–KCl–$Na_2S$ melt. Potentiostatic and galvanostatic electrolyses were conducted using the three-electrode set-up described above. All electrochemical measurements were conducted by an electrochemical workstation (Auto lab PGSTAT302N, Metrohom AG), and the potential between the working and counter electrodes was monitored by a four-electrode battery testing system (Maccor 4300). Throughout this manuscript, all potential values are expressed relative to $Na^+/Na$ (−2.2 V versus Ag/AgCl) unless otherwise noted. When the electrolysis was terminated, the electrodes were lifted out of the melt and cooled down in the argon protected headspace of the stainless steel test vessel. The working electrode graphite cup was then cross-sectioned and characterized by X-ray diffraction (Panalytical X'pert Pro Multipurpose Diffractiometer with Cu Kα radiation) and scanning electron microscopy (JEOL 6610LV) fitted with energy dispersive spectrometer (EDS, IXRF system, Model 55i).

**Two-electrode fused quartz cell.** A demonstration cell made of a closed-one-end fused quartz tube (ID 1.5 cm) was used to observe the evolution of the gas product from the anode. A unit of 3 g $Sb_2S_3$ together with 15 g NaCl–KCl–$Na_2S$ were introduced into the fused quartz cell to be heated by natural gas and oxygen flame. Two graphite rods of diameter 3 and 6 mm served as cathode and anode, respectively. Galvanostatic electrolysis at 500 mA cm$^{-2}$ was conducted between the two graphite electrodes. During the electrolysis, the demo cell was under argon flow.

**Data availability.** The authors declare that the data supporting the findings of this study are available within the article and its Supplementary Information files.

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

## Acknowledgements

This research was funded by US Department of Energy, Advanced Research Projects Agency-Energy (Award No. DE-AR0000047) and Total, S.A. We thank Prof Antoine Allanore for valuable discussions and Prof Allen J. Bard for valuable discussions and for allowing us to construct the fused quartz demonstration cell in his lab.

## Author contributions

H.Y. and D.R.S. conceived the idea of this research; and H.Y. and B.C. performed the experiments and analysed the results; H.Y.Y. and B.C. wrote the original draft of the manuscript; D.R.S. substantially revised, edited and submitted it.

## Additional information

**Competing financial interests:** The authors declare no competing financial interests.

