## [Peer Review File · Nature Communications]

REVIEWERS' COMMENTS: (reviewer numbers are consistent with those used during peer review at Nature Materials)

Reviewer #1 (Remarks to the Author):

I have looked through the revised paper and most of the issues have been addressed. The two outstanding points are:

1. The authors need to state somewhere that the process does not refine the antimony so if you feed an impure stibnite, the cathodic product will also be impure.
2. Lines 133/134. This gives the impression that this route is much better than the process for the extraction of aluminium. The main difference between the two kWh/kg is due to the fact that the atomic weight of Sb is 122g compared to 27g for aluminium so for the same amount of current five times as much Sb is formed.

As I said in my original review, it is a very good paper but if the intention is to draw the work to the attention of the metallurgical community, it would be better to publish it in Metallurgical Transactions rather than Nature Communications but then, on the other hand, Nature Communications is far more prestigious!

Reviewer #3 (Remarks to the Author):

The paper deals with an interesting topic, and it is related to developing a novel method for extracting metals. The results should be interesting for the research community within the field of molten salt electrochemistry and metal production.

The paper should be published.

I would like to ask the authors to consider the following comments.

It is said on page 6 that at 500 mA/cm² the energy consumption is 1.5 kWh/kg Sb and the current efficiency is 88 %. It's not "fair" to compare with the Al electrolysis process due to the large difference in atomic weight. The current efficiency is actually rather low at 88 %. I don't find a lot of explanation for the loss in current efficiency. The cell voltage is very low, but it seems like (Fig. 4) that the ohmic voltage drop in the electrolyte is not included in the cell voltage. In the Al electrolysis process and other molten salt processes this is the most important part of the cell voltage.

Reviewer #5 (Remarks to the Author):

This paper demonstrates a novel approach to produce transition metals by direct electrolysis of their sulfides. Specifically, the use of a second-phase ionic electrolyte mitigates electronic conduction in the cell, which has previously stymied efficient high-volume production of transition metals by electrolysis. This seminal paper should usher in a new environmentally-sound methodology for extraction of metals from sulfide ores.

REVIEWERS' COMMENTS: (reviewer numbers are consistent with those used during peer review at Nature Materials)

Reviewer #1 (Remarks to the Author):

I have looked through the revised paper and most of the issues have been addressed. The two outstanding points are:

1. The authors need to state somewhere that the process does not refine the antimony so if you feed an impure stibnite, the cathodic product will also be impure.

Agreed. Manuscript has been revised accordingly.

2. Lines 133/134. This gives the impression that this route is much better than the process for the extraction of aluminium. The main difference between the two kWh/kg is due to the fact that the atomic weight of Sb is 122g compared to 27g for aluminium so for the same amount of current five times as much Sb is formed.

Agreed. Manuscript has been revised accordingly. The reviewer correctly points out the difference in atomic masses. On further reflection we recognized that the productivity difference is also a reflection of the relative stabilities of the feedstock, i.e., Al_2O_3 is much more difficult to reduce than Sb_2S_3 which means that an Al cell must necessarily operate at a higher voltage. It was a sloppy comparison as originally written. Thank you for drawing attention to this.

As I said in my original review, it is a very good paper but if the intention is to draw the work to the attention of the metallurgical community, it would be better to publish it in Metallurgical Transactions rather than Nature Communications but then, on the other hand, Nature Communications is far more prestigious!

Agreed. Nature Communications is *far, far more prestigious*.

Reviewer #3 (Remarks to the Author):

The paper deals with an interesting topic, and it is related to developing a novel method for extracting metals. The results should be interesting for the research community within the field of molten salt electrochemistry and metal production.

The paper should be published.

I would like to ask the authors to consider the following comments.

It is said on page 6 that at 500 mA/cm² the energy consumption is 1.5 kWh/kg Sb and the current efficiency is 88 %. It's not "fair" to compare with the Al electrolysis process due to the large difference in atomic weight. The current efficiency is actually rather low at 88 %. I don't find a lot of explanation for the loss in current efficiency. The cell voltage is very low, but it seems like (Fig. 4) that the ohmic voltage drop in the electrolyte is not included in the cell voltage. In the Al electrolysis process and other molten salt processes this is the most important part of the cell voltage.

Agreed. Please see above for the response on the comparison with Al electrolysis.

As for loss of current efficiency, we did not focus attention on possible mechanisms as they are to a great extent a function of cell design. We know from lab-scale Al cells that the measurement of current efficiency is not representative of the performance in an industrial cell. Our point here is simply that at 88% the current efficiency is in the range of acceptability. Twenty years ago Al cells ran at 90% current efficiency and the Dow cell for the electrolytic production of Mg ran at 80% current efficiency. Both technologies were profitable in their day.

On the matter of cell voltage the reviewer makes a good point here. Our cell voltage does indeed include the ohmic drop across the electrolyte. It is true that in Al electrolysis this is the highest component of the cell voltage *at industrial scale, i.e. self heated*. The lab cell is externally heated and has a rather different geometry so the breakdown of cell voltage can be quite different from what one sees in the industrial cell of the same chemistry.

Reviewer #5 (Remarks to the Author):

This paper demonstrates a novel approach to produce transition metals by direct electrolysis of their sulfides. Specifically, the use of a second-phase ionic electrolyte mitigates electronic conduction in the cell, which has previously stymied efficient high-volume production of transition metals by electrolysis. This seminal paper should usher in a new environmentally-sound methodology for extraction of metals from sulfide ores.

Many thanks for the kind words.